# Antioxidants versus Food Antioxidant Additives and Food Preservatives

**DOI:** 10.3390/antiox8110542

**Published:** 2019-11-11

**Authors:** Rafael Franco, Gemma Navarro, Eva Martínez-Pinilla

**Affiliations:** 1Chemistry School, University of Barcelona, 08028 Barcelona, Spain; 2Centro de Investigación Biomédica en Red Enfermedades Neurodegenerativas (CiberNed), Instituto de Salud Carlos III, 28031 Madrid, Spain; 3Department of Biochemistry and Physiology, Faculty of Pharmacy, University of Barcelona, 02028 Barcelona, Spain; 4Departamento de Morfología y Biología Celular, Facultad de Medicina, Universidad de Oviedo, 33006 Oviedo, Spain; martinezpinillaeva@gmail.com; 5Instituto de Neurociencias del Principado de Asturias (INEUROPA), 33003 Oviedo, Spain; 6Instituto de Investigación Sanitaria del Principado de Asturias (ISPA), 33011 Oviedo, Spain

**Keywords:** human safety, food rotting, food decay, food contamination, REDOX reaction, taste, sulfites

## Abstract

Natural and processed foods are fragile and can become unpalatable and/or rotten. The processed food industry uses preservatives to enable distribution, even to different continents, and to extend the useful life of their products. Preservatives impede oxidation, a mandatory step in rotting, either by aerobic or anaerobic mechanisms. From a functional point of view, these compounds are antioxidants, and, therefore, a kind of contradiction exists when a preservative is considered “bad” for human health while also thinking that antioxidants provide benefits. The basis of antioxidant action, the doses required for preservation, and the overall antioxidant action are revisited in this work. Finally, the bad and the good of food additives/preservatives are presented, taking into account the main mediator of antioxidant beneficial actions, namely the innate mechanisms of detoxification. Foods that strengthen such innate mechanisms are also presented.

## 1. Introduction

Approved food additives, denoted in Europe by an “E” followed by, at least, three numbers, are divided into different categories. Two of them are considered in this paper: “antioxidants” and “preservatives.” Food preservation in the past mainly relied on cold temperatures (achieved using snow), sodium chloride (salt), sugar (bioavailable sugars at a high concentration, as in honey or jam), spices (cumin or cinnamon) or vegetable oils, none of which cause the concern of users [1,2]. From the very beginning of the sale of processed foods in shops/markets, there has been controversy regarding whether food additives are harmful, i.e., whether they are detrimental for human health. The debate is on stage, and the message is, fortunately, that additives are needed and for good reasons, some of which are not of general knowledge. Additives in the amounts ingested by humans are safe or GRAS (generally recognized as safe) in the standard nomenclature [3]. This paper does not consider other additives such as pesticides that cannot be considered safe and that lack specific regulation on final-product labeling.

## 2. Preservatives Versus Antioxidants

First of all, it is worth noting that there is no a clear border for a given drug being a preservative or antioxidant. The term preservative is more user-friendly and more direct, as any substance that prevents food rotting is immediately considered a preservative. A prototypical preservative is the enzyme lysozyme. On the one hand, it is found in egg whites but also in tears to avoid eye bacterial infection. On the other hand, it acts as bactericide because it cleaves the carbohydrates of bacteria wall(s). Another example is sorbic acid, or sorbates, which is efficacious because there are few microorganisms that can use it as a source of energy, thus preventing the starting of cell growth; accordingly, these compounds are kind of bacteriostatic [4,5]. The definition of an antioxidant is not so straightforward, as one can use a rigorous chemistry path, i.e., any substance that is a reductant and reacts with an oxidant, or a more lax definition such as one that may be found in the Britannica Encyclopedia: “Antioxidant: any of various chemical compounds added to certain foods, natural and synthetic rubbers, gasolines, and other substances to retard autoxidation, the process by which these substances combine with oxygen in the air at room temperature. Retarding autoxidation delays the appearance of such undesirable qualities as rancidity in foods, loss of elasticity in rubbers, and formation of gums in gasolines. Antioxidants most commonly used are such organic compounds as aromatic amines, phenols, and aminophenols.” In the former, the term is relative, as one given substance may be both oxidant or antioxidant depending on which is the “other” reacting molecule. This does not mean as any substance participating in a reduction–oxidation (REDOX) reaction has the so-called standard reduction potential (ε^0^), which can be calculated (see for [6,7,8] for pioneering studies on technical methods for ε^0^ calculation and [9] as a recent article on this subject). Nevertheless, as much as possible, we avoid to use ε and other markedly technical chemical terms in this paper. To summarize, in the food industry field preservatives are used to avoid rotting while antioxidants are used to prevent the chemical (oxidization) reactions leading to unpleasant taste and/or smell.

From a strict chemistry view, in the list of preservatives and antioxidants displayed in Table 1, there are pure antioxidants, pure preservatives, and compounds with mixed properties. Ascorbic acid (vitamin C) is a pure antioxidant, whereas sorbic acid (or sorbate salts) and benzoic acid (or benzoate salts) are pure preservatives. However, it is not completely clear why propyl gallate is considered an antioxidant or sodium sulfite is considered a preservative. Sulfites and bisulfites may be considered in vivo antioxidants: sulfites are oxidized to sulfates, and bisulfites, also in the preservative list, are oxidized to bisulfates [10]. In other words, their mode of action in food preservation is that of antioxidants. It is true that sulfites may be involved in other actions, one of them being the prevention of undesired enzymatic reactions in foods/beverages [10,11,12]. At this point, it is interesting to discuss toxicity data. In this sense, we argue that dosage must be considered and that not all antioxidants are innocuous, but food additives at the doses used can be considered safe. Surely, this data-based opinion can be extended to food preservatives.

The toxicity of compounds in Table 1 is fairly low. An example retrieved from the Toxnet platform in October, 2019 [14] showed that 4.9 g/kg is the 50% per oral lethal dose (LD_50_) of potassium sorbate in rodents. As seen on fcsimage [15], also in October, 2019, the LD_50_ value for the allegedly dangerous ethylene glycol is 4.4 g/kg in rats and 100 mL of pure compound in humans. Overall, these data indicate that the additives in Table 1 should be considered generally safe at the doses found in final food products. Exceptions may occur in the case of poor control quality or in cases of acute exposure such as that reported for sulfur dioxide (see below). We have experience in preservatives safety; a pre-graduate student in her last academic year fainted after ingesting a tiny amount of a sodium azide-diluted solution, which is used to prevent bacterial growth in biomolecule-containing preparations. After investigating our first hypothesis regarding possible poisoning, we found out that toxicity of sodium azide for mammals is low and that this compound has been reported as a potential antihypertensive drug [16,17]. In summary, the amount taken in a meal is too low to consider that food preservatives/antioxidant additives are dangerous in the short or long run.

Similarly, there is a consensus regarding the health benefits of moderate wine consumption despite that the fact that wine contains sulfites. What is more toxic in wine, sulfites or alcohol? In moderate amounts, both are safe; however, putting aside the potential addictive effect of ethanol, excess wine consumption may lead to cirrhosis due to ethanol, but no serious detrimental effect has been attributed to sulfites [18,19]. Long-term scientific experience with sulfite-containing wine has shown that prejudices regarding wine come from ethanol and may come from pesticides in grapes but not from approved wine additives.

## 3. Precautions with Preservatives and Antioxidant Food Additives

One of the main issues surrounding food additives is the lack of information about their contents in a given product. Therefore, the safety of food additives has been assumed in a semi-blind basis, i.e., knowing the chemical structure but lacking the concentration in the final product and/or in the final meal/beverage [3]. 

As a first example, we may consider that sulfites lead to the degradation of vitamin B1 (thiamine), which is an adverse effect; obviously, this can be conveniently addressed if enough vitamin B1 is taken in the diet or as a dietary supplement (see [20]). Sulfite, bisulfite and metabisulfite additives have been quantified in the food of a Korean population for which the “acceptable daily intake” (ADI) was assessed. The authors of that study concluded that “The overall exposure to antioxidants and color fixatives in foods did not exceed the ADI. We estimated exposure to be within safe levels for all population groups” [21]. For its part, sulfur dioxide can be dangerous depending on the dose, and there is a situation that illustrates the precautions needed for some additives and/or in some circumstances. It is known that the compound irritates the respiratory tract, and it is, thus doubtful that it may cause harm unless inhaled. However, a closer look has indicated that it may cause serious problems to certain individuals, such as those suffering from asthma. To our knowledge, the first reported case of such a reaction was due to dried fruits treated with the gas and distributed in a tightly sealed package. The sulfur dioxide coming out from the plastic bag, when opened, was enough to produce acute episodes in an asthma patient; authors reported: “coughing, shortness of breath, expiratory wheezes and rhonchi” [22]. The allergenic effect of sulfur dioxide can be likely ruled out, but we cannot rule out the irritant properties that could not affect healthy subjects but could affect patients with respiratory diseases. In spite of this, it must be highlighted that, being a gas, the side effects of sulfur dioxide can be minimized if the food is aerated (or heated in open pans) before consumption ([22] and face-to-face comment by Paul F. Wehrle).

Another example of precaution advice is related to sodium metabisulfite (labeled as E223; Table 1), or other additives in shrimps, prawns or similar animals (even if directly frozen). The “head” of prawns quickly becomes dark (browning) due to polyphenol oxidase-induced melanosis [23,24], and producers know that consumers prefer that these crustaceans keep the red color. Though sucking fresh crustaceans is pleasant to many and safe, if the product has been hypothetically treated with unknown quantities of additives, such as sodium metabisulfite, a similar action could be neither tasty nor completely safe (if the dose is high or a high number of pieces are consumed). An advice in cases in which plenty of prawns are going to be cooked and consumed is to remove the “head,” which retains more chemicals than the rest of the body. A possible alternative is the use of resorcinol derivatives that may have equal benefits in crustaceans but cause less concern than bisulfites [25].

In the last two years in Spanish shops, we have noticed a change in the way fresh tuna fish is sold. Years ago, tuna fish was red when fresh and became brown (oxidized, starting by the surface) with time. By looking at the color after cutting a steak, one may figure out how fresh it is. Additionally, the shops got rid of fresh tuna fish quickly, i.e., the same day. Nowadays, tuna is stored in a plastic wrap and keeps its reddish color for days. It seems that this product has been treated with an antioxidant that, at present, is not labeled. The quantity of additives to keep the red color may be quite high, and, again, no quality control may be performed (or, if performed, may not be in compliance with good laboratory practices). In parallel, we have noticed that the label of the packs of “fresh” fish products is not entirely clear. Apart from altering the taste of the real food, procedures that are not well controlled do not guarantee safety even if a generally safe additive is used (we mean that the concentration is surely quite high and the additive is going to be ingested in full with the food). In our own experience with sausages that have not been cooked but have been cured using additives, it seems that producer (small producer usually) uses an excess of the additive in some cases. This situation, which may be due to lack of appropriate quality control, can be detected when the taste buds notice roughness and/or dryness upon sausage biting.

Not long ago, a well-known and successful Spanish brand of dairy products publicized the benefits of its new yogurts by stating that they were 100% natural and free from “artificial” additives. However, this advertising campaign caused a great debate in the public opinion, since it was discovered that their products contained E960 (steviol glycosides) whose extraction from *Stevia rebaudiana* is performed in a laboratory using solvents and ion exchange resins that, in other countries, do not allow for labeling as a “natural ingredient” [26]. Moreover, the specific product, a yogurt, also contained E162 (beetroot red), a colorant of natural origin but denoted in the label as a betanin pigment, for which no adequate toxicity studies were available. The brand was forced to rectify the labeling days later, but they claimed that their additives were natural while those of other brands were not [27]. This is one of the many examples of considering a natural product better than a non-natural one. 

In summary, despite the fact that the general population prefers “natural” compounds/additives, in food and in other fields, natural versus non-natural is not a convenient approach to distinguish between friend and foe. Proper discussion and convenient examples are provided in the book by Collman [28], whose subtitle is “Surprising facts about food, health and environment;” the author shows that not all products made by nature are safe, nor are man-made products unsafe. Additionally, one must be aware when noticing something looking bizarre via taste or smell senses, or when a lot of processed food is being consumed in a unique intake: the “dose” matters.

## 4. Antioxidants: Beneficial or Detrimental for Human Health

This section deals with the interesting debate that de facto exists between the defendants of the goodness of antioxidants and the persons worried by antioxidants as food additives. From a strict point of view, there are no good and bad antioxidants; actually, it is not convenient to raise a debate in terms of good and bad substances. Often, we use water as an example of something that is neither good nor bad. It is needed for life on Earth, but one must use it correctly; pure water usage has the same deadly consequences as using sea water. Water must be consumed with the right concentration of water itself and with the right composition and concentration of salts. Another example of the “danger” of water was reported in Legends website ([29], as quoted in [28]) in which the statement “Accidental inhalation can kill you” together with other complementary information led 43 out of 50 recipients to be convinced that the substance (water) should be banned. Analogously, the interest of the academic researchers and/or entrepreneurs in placing the focus in the health benefits of any antioxidant makes little sense. On the one hand, “antioxidant” is a relative term that depends on the reaction with a reductant. If we go to an extreme, some of the most potent antioxidants, even in vivo, are those containing sulfhydryl (—SH) groups. Accordingly, one could assume that SH_2_ (hydrogen sulfide), which is an antioxidant, may provide benefits. However, SH_2_ is one of the most toxic gases, more toxic than a more well-known chemical: hydrogen cyanide (liquid but with very poisonous vapors; its boiling point is <30°). On the other hand, any antioxidant action measured in vitro or very indirectly in vivo may not have physiological relevance, either by lack of appropriate reductant or by a kinetic issue, e.g. slow reaction. The described variety of methods for testing antioxidant power (see [30] for details) faces the wall that antioxidant power is a constant in standard conditions (reduction potential; ε^0^) and, unfortunately, none (it may be some exceptions) of the methods now used take this fact into account. In addition, usually REDOX reactions take place mole by mole (eventually with 1:2 or 2:1 stoichiometry); therefore, no antioxidant could prevent the number of oxidants that are produced in a human body. It should be noted that a good proposal to determine overall REDOX status has not been yet considered in a routine basis. This method consists of measuring the ratio of oxidized versus reduced coenzyme Q, i.e., of ubiquinone versus ubiquinol, in plasma [31,32]. Surely, such an in vivo measured ratio will be more appropriate than measuring the total level of the “antioxidant;” in addition, an increased oxidized versus reduced status would indicate whether the compound is really behaving as in vivo antioxidant or not.

Once more, the belief that natural substances—food additives in the context of this paper—are “better” than synthetic molecules has not been substantiated by scientific data [28]. Natural polyphenols may be beneficial or detrimental, i.e., they may be good or bad [33,34]. The reasons for this are basically those provided in the previous paragraph, but a further detail is as follows: there is no risk at the concentrations present in food. In other words, to assume that all polyphenols have health benefits because they act as antioxidants in vitro is not a solid reason to sell them as supplements with health benefits [34,35]. Taken together, this debate leads to the conclusion that the antioxidants that are used as food additives do not pose any risk to the health at the concentrations approved/suggested. All of the antioxidant food additives, whose nomenclature in Europe is E plus a number (EXXX), have been carefully studied to conclude that, at the right, doses they are GRAS. 

We would like to further highlight a theme whose relevance has not yet been generally accepted. We refer to the need to boost the innate mechanisms of defense in mammals. This consists of fueling, for instance, the mechanism that keeps Fe^2+^ in hemoglobin; if the ion is oxidized to Fe^3+^, the resulting molecule, methemoglobin, is unable to bind (and transport) oxygen. The Fe^3+^/Fe^2+^ reduction mechanism operating in the erythrocyte does not rely in antioxidant supplements. It is well known and consists of cycles of oxidized/reduced glutathione (a cysteine-SH-containing tripeptide) that needs reduced nicotinamide adenine dinucleotide phosphate (NADPH), produced from the reducing power in glucose and glucose-6-phosphate dehydrogenase. Hemolytic anemia is a well-known inherited disease due to the reduced expression/activity of glucose-6-phosphate dehydrogenase [36,37]. We argued in a previous paper that substances from foods, from additives or from supplements, that fuel those innate mechanisms are real antioxidants [35]. Obviously, not all in vitro antioxidants are able to afford direct actions on innate detoxification mechanisms. 

There is an additional aspect to consider that is whether the opposite to an antioxidant, i.e., an oxidant or a pro-oxidant, may be of benefit. The answer is “yes.” Some foods, the most known being fava beans, contain pro-oxidants that serve to keep the antioxidants functional and keep their mechanisms ready for operating in the erythrocyte. Vicine and convicine are the key compounds that lead to hemolysis in the absence of the enzyme [38]. By the same token, additives that are considered “less good” may provide significant benefits if they contribute to keeping the innate antioxidant/detoxification mechanisms alive and ready, i.e., with all the components at adequate levels of expression. Regarding sulfite-containing food additives, it has been shown that these substances “are relatively harmless because animals have efficient detoxication systems that oxidize sulfite to sulfate” [11,12,39]. It is therefore tempting to speculate that sulfites in wine and in other foods serve to keep the human body ready to cope with some of the environmental factors impacting human life. An introduction to these so-called hormetic mechanisms and references to first reports on hormesis can be found elsewhere [33,34,40,41]. It is worth mentioning that genes, whose products alter their expression level depending on environmental factors, are known as vitagenes [42].

## 5. Future Prospects on Health Concerns Due to Food Additives

Preservatives and antioxidant food additives are safe if used with caution and follow the approved rules. It is, however, important to know whether these compounds can have direct benefits on the human body and do not only preserve foods from oxidation or rotting. We think that the pathways by which sulfites are metabolized and eliminated should be characterized while looking into their possible benefits, i.e., whether the human body is readier to cope with external stress when consuming those additives. This may include the determination of expression levels of “detoxifying” enzymes when a mammal eats food with or without additives. It is predicted that hormesis and hormetic mechanisms related to food and food additives will be active research fields that may provide valuable answers and sound food-based interventions.

Interestingly, the relationships between glutathione and ascorbic acid (vitamin C) have been described in plants for both correct metabolism and stress tolerance [43]. Though we have argued that vitamins do not act as antioxidants, i.e., the physiological role of vitamins is not related to any global antioxidant activity [44], ascorbate synthesis in liver mice needs glutathione [45]. In this sense, it could be hypothesized that vitamin C, in humans, is somehow related to glutathione-based innate detox mechanisms and, eventually, to other not-yet-characterized antioxidant machineries. This issue is worth being considered by scientists in the antioxidant research field.

We would like to end this final section with a note of caution based in the images in Figure 1. Is more beneficial to use enough additives (in terms of type and concentration) to prevent all “senescence/decay” of aliments or to consume food that has not expired but that shows signs of contamination by living organisms that are contaminating it? The identity of those contaminants, probably yeasts/fungi in the muffins of Figure 1 is not yet known, and, therefore, their effects on human health are not known either.

## Figures and Tables

**Figure 1 antioxidants-08-00542-f001:**
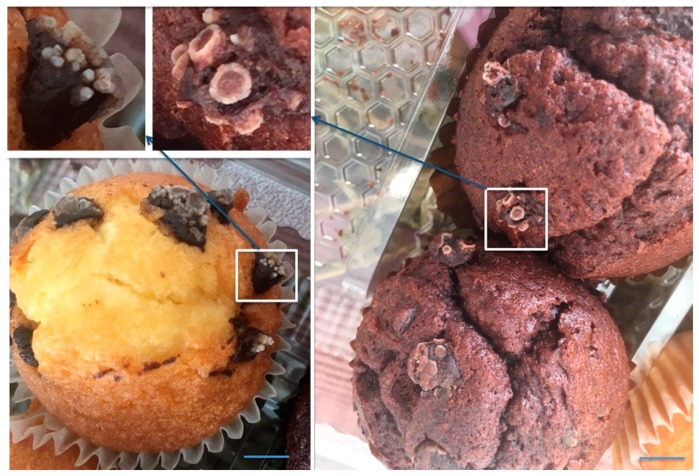
**Images of muffins with contaminants that arose way before the caducity date**. Images from muffins in which all the chocolate chips presented this aspect two-to-three days after package opening and more than 45 days before the deadline written in the label. Magnification images of a chip in brown and of a chip in white muffins (insets) show signs of biological contamination. The product comes from our own production of a supranational chain of big supermarkets and was bought in a town in Catalonia, Spain. Spanish health authorities were informed, but no response, on any issue, has been yet given after four months. The label in the package indicated that, apart from unspecified “aromas,” the following additives were present: E160a(iii) (β-carotene), E170 (calcium carbonate) and E516 (calcium sulfate) in white muffins; E412 (guar gum), E466 (sodium carboxymethyl cellulose), E481 (sodium stearoyl lactate) and E500 (sodium carbonates) in brown muffins; and E200 (sorbic acid), E202 (potassium sorbate), E450 (diphosphates) and E471 (mono- and diglycerides of fatty acids) in both types of muffins. Scale bar: approximately 10 mm.

**Table 1 antioxidants-08-00542-t001:** Food preservatives and antioxidants.

Antioxidants
E300	Ascorbic acid	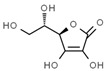	E310	Propyl gallate	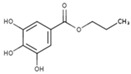
E301	Sodium ascorbate	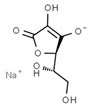	E315	Erythorbic acid	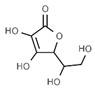
E302	Calcium ascorbate	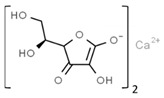	E316	Sodium erythorbate	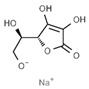
E304	Fatty acid esters of ascorbic acid	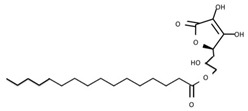	E319	Tertiary-butyl hydroquinone (TBHQ)	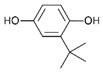
E306	Tocopherols	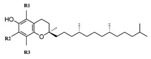	E320	Butylated hydroxyanisole (BHA)	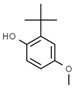
E307	Alpha-tocopherol	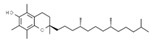	E321	Butylated hydroxytoluene (BHT)	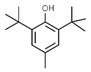
E308	Gamma-tocopherol	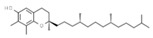	E392	Extracts of rosemary	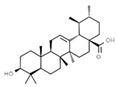
E309	Delta-tocopherol	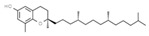	E586	4-Hexylresorcinol	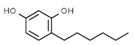
Preservatives
E200	Sorbic acid	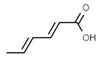	E228	Potassium hydrogen sulfite	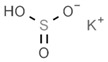
E202	Potassium sorbate	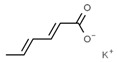	E234	Nisin ^a^	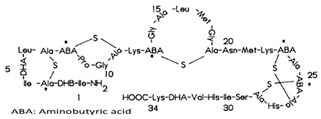
E210	Benzoic acid	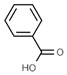	E235	Natamycin	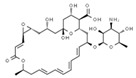
E211	Sodium benzoate	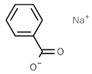	E239	Hexamethylene tetramine	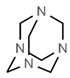
E212	Potassium benzoate	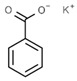	E242	Dimethyl dicarbonate	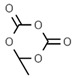
E213	Calcium benzoate	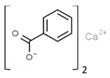	E243	Ethyl lauroyl arginate	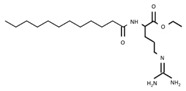
E214	Ethyl *p*-hydroxybenzoate	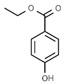	E249	Potassium nitrite	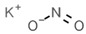
E215	Sodium ethyl *p*-hydroxybenzoate	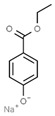	E250	Sodium nitrite	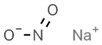
E218	Methyl *p*-hydroxybenzoate	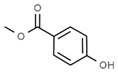	E251	Sodium nitrate	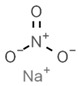
E219	Sodium methyl *p*-hydroxybenzoate	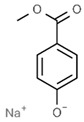	E252	Potassium nitrate	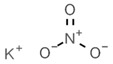
E220	Sulphur dioxide	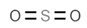	E280	Propionic acid	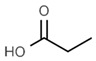
E221	Sodium sulfite	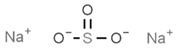	E281	Sodium propionate	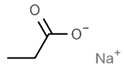
E222	Sodium hydrogen sulfite	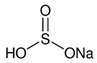	E282	Calcium propionate	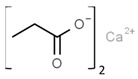
E223	Sodium metabisulfite	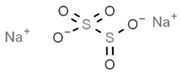	E283	Potassium propionate	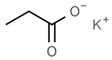
E224	Potassium metabisulfite	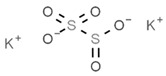	E284	Boric acid	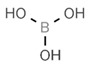
E226	Calcium sulfite	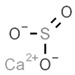	E285	Sodium tetraborate; borax	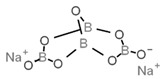
E227	Calcium hydrogen sulfite	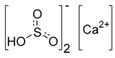	E1105	Lysozyme (*N*-acetylmuramide glycanhydrolase)	Enzyme de 14 kilodaltons

^a^ Nomenclature of Liu and Hansen, 1990 [13].

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
