# Peer review of "Antioxidants versus Food Antioxidant Additives and Food Preservatives"

_antioxidants, 2019, doi:10.3390/antiox8110542_

Round 1

Reviewer 1 Report

This review work provides an expert overview of an antioxidants in foods.

This review is quite complete, well-ordered in its form and well written.

The discussion is very synthetic, insisting on the most interesting aspects of this review allowing to put into perspective the most recent works in order to clarify them or to find them future applications. However, it would be interesting to insist a little more on the use of natural antioxidants or preservatives as opposed to synthetics, in particular as future prospects.

Considering all of these elements, I suggest a minor revision of this work.

Author Response

The discussion is very synthetic, insisting on the most interesting aspects of this review allowing to put into perspective the most recent works in order to clarify them or to find them future applications. However, it would be interesting to insist a little more on the use of natural antioxidants or preservatives as opposed to synthetics, in particular as future prospects.

Answer: we appreciate the comments and suggestion, which has been taken into account in the revised version of the manuscript. As the reviewer proposed we have rewritten some parts of the paper in order to deepen in this topic.

Observation: major changes are highlighted in yellow. In addition, a deep English grammatical revision was performed in the new version of the paper in order to facilitate its proper reading and understanding. Grammatical corrections are not highlighted.

Reviewer 2 Report

This is a very interesting and critical review of a "social concern" topic addressing antioxidants versus food antioxidant additives and food preservatives. In my opinion, this manuscript is acceptable for publication in Antioxidants as it will be interesting not only for the scientific community but also for the society in general, who is having the main concerns against the use of "added" food antioxidants and preservatives, mainly due to the lack of correct information.

My only suggestion to the authors is that they also consider at some point that one of the main reasons for the society to be reluctant on the use of "food additives" and "food preservatives" is not only due to the "possible" toxic effect of a compound (many people consider that a chemical is bad only because is a chemical), but to the fact that many people prefer natural chemical products agains synthetic chemical products. At the end, the effect on the final food will be the same, because is the same chemical, but society will always prefer something that was obtained from natural products than something synthesized. I suggest the authors to comment that topic on the manuscript. Again, it will always be caused by a lack of information of the product labeling.

I have to say that I liked a lot when authors were giving their critical opinion on some aspect, but not when this opinion seems to lack real evidence. In those cases I will suggest the authors to try to say the same but in a more correct way. For example, when the authors are addressing in lines 129-143 the fresh tuna case in Spanish shops, I believe that saying: "Undoubtedly this product has been treated with an antioxidant that, at present, is not labeled" is a very strong affirmation (that probably is true), but without any reported evidence. So, try to rewritte this affirmation. It must be clear that this is only the authors opinion.

Some minor English errors need to be corrected.

Author Response

This is a very interesting and critical review of a "social concern" topic addressing antioxidants versus food antioxidant additives and food preservatives. In my opinion, this manuscript is acceptable for publication in Antioxidants as it will be interesting not only for the scientific community but also for the society in general, who is having the main concerns against the use of "added" food antioxidants and preservatives, mainly due to the lack of correct information.

Answer: thanks for the positive comment.

My only suggestion to the authors is that they also consider at some point that one of the main reasons for the society to be reluctant on the use of "food additives" and "food preservatives" is not only due to the "possible" toxic effect of a compound (many people consider that a chemical is bad only because is a chemical), but to the fact that many people prefer natural chemical products agains synthetic chemical products. At the end, the effect on the final food will be the same, because is the same chemical, but society will always prefer something that was obtained from natural products than something synthesized. I suggest the authors to comment that topic on the manuscript. Again, it will always be caused by a lack of information of the product labeling.

Answer: we appreciate the suggestion that has been taken into account in the revised version of the manuscript. As the reviewer proposed we have rewritten some parts of the paper in order to deepen in this topic.

I have to say that I liked a lot when authors were giving their critical opinion on some aspect, but not when this opinion seems to lack real evidence. In those cases I will suggest the authors to try to say the same but in a more correct way. For example, when the authors are addressing in lines 129-143 the fresh tuna case in Spanish shops, I believe that saying: "Undoubtedly this product has been treated with an antioxidant that, at present, is not labeled" is a very strong affirmation (that probably is true), but without any reported evidence. So, try to rewritte this affirmation. It must be clear that this is only the authors opinion.

Answer: we appreciate the comment, which is considered in the revised version of the manuscript.

Observation: major changes are highlighted in yellow. In addition, a deep English grammatical revision was performed in the new version of the paper in order to facilitate its proper reading and understanding. Grammatical corrections are not highlighted.